# Treatment Prognosis of Restored Teeth with Crown Lengthening vs. Deep Margin Elevation: A Systematic Review

**DOI:** 10.3390/ma14216733

**Published:** 2021-11-08

**Authors:** Maryam H. Mugri, Mohammed E. Sayed, Binoy Mathews Nedumgottil, Shilpa Bhandi, A. Thirumal Raj, Luca Testarelli, Zohaib Khurshid, Saurabh Jain, Shankargouda Patil

**Affiliations:** 1Department of Maxillofacial Surgery and Diagnostic Sciences, College of Dentistry, Jazan University, Jazan 45142, Saudi Arabia; dr.mugri@gmail.com; 2Department of Prosthetic Dental Sciences, College of Dentistry, Jazan University, Jazan 45142, Saudi Arabia; drsayed203@gmail.com (M.E.S.); drsaurabhjain79@gmail.com (S.J.); 3Department of Prosthodontics and Dental Implantology, College of Dentistry, King Faisal University, Al-Ahsa 31982, Saudi Arabia; bmuottil@kfu.edu.sa (B.M.N.); drzohaibkhurshid@gmail.com (Z.K.); 4Department of Restorative Dental Science, Division of Operative Dentistry, College of Dentistry, Jazan University, Jazan 45142, Saudi Arabia; shilpa.bhandi@gmail.com; 5Department of Oral Pathology and Microbiology, Sri Venkateswara Dental College and Hospital, Chennai 600130, India; thirumalraj666@gmail.com; 6Department of Oral and Maxillo-Facial Sciences, Università di Roma La Sapienza, 00185 Roma, Italy; luca.testarelli@uniroma1.it; 7Department of Maxillofacial Surgery and Diagnostic Science, Division of Oral Pathology, College of Dentistry, Jazan University, Jazan 45142, Saudi Arabia

**Keywords:** crown lengthening, deep margin elevation, tooth decay, restoration, systematic review

## Abstract

Crown lengthening surgery and deep margin elevation are two distinct approaches used to manage decayed teeth. This systematic review examined the survival rate of badly decayed teeth when restored using the crown lengthening technique and compared it to the deep margin elevation technique. The search was conducted during July 2020 and then again updated at the end of July 2021, and no restriction concerning publication status and time was applied during the search. Cochrane Database, EBSCO, Scopus, and Medline databases were searched electronically for relevant literature. Google Scholar was used as a secondary source. Predefined inclusion and exclusion criteria were used to select the relevant articles. PRISMA guidelines were followed. The focused PICO question was: ‘Does the crown lengthening technique (I) provide a better survival rate (O) than deep margin elevation technique (C) following the restoration of badly decayed teeth (P).’ A total of six articles were included after performing screening based on the eligibility criteria. Four studies focused on crown lengthening while two focused on deep margin elevation technique. A majority of the studies showed a high risk of bias owing to methodological insufficiencies. Crown lengthening (CL) treated cases showed a change in the free gingival margin at six months post-surgery. A tissue rebound was seen that was correlated to the periodontal biotype. Teeth treated with the deep margin elevation (DME) technique showed high survivability. There is a lack of high-quality trials examining surgical comparisons between CL and DME with long-term follow-up. Patient- and dentist-reported outcomes have not been given adequate consideration in the literature. Based on the limited evidence, it can be concluded that for restorative purposes, crown lengthening surgery can be successful in long-term retention of restored teeth. However, the deep margin elevation technique has a better survival ratio. Future well-designed and executed research will have an effect on the evidence and level of certainty for the best approach to treating severely decayed teeth.

## 1. Introduction

The replacement of large Class II restorations can result in subgingival interdental margins. The use of direct adhesives with shrinkage stress-reduction techniques is not ideal in large defects. Post-curing effects that continue for several days after composite resin placement render the dentin gingival seal unsecured [1]. Considering the size of these large defects, onlays/inlays fabricated using chairside computer-aided design/computer-assisted manufacturing (CAD/CAM) would be a superior alternative [2].

Apical displacement of supporting tissues during surgery exposes gingival margins, resulting in anatomical complications and loss of attachment in cases of root proximity, furcation, and concavities. It is difficult to maintain and generate changes in the gingival margins once they are exposed to the oral environment [3]. The deep margin elevation (DME) technique elevates the cervical margin of a subgingival restoration by placing composite resin. This is achieved following the matrix placement under the rubber dam isolation. DME improves the bond and marginal seal of indirect adhesive restorations and results in immediate dentin sealing [4]. The adhesive composite resin base is used for reinforcing the undermined cusps, providing necessary geometry for only/inlay restorations, sealing the dentin, and filling undercuts along with supragingival elevation of margin.

Surgical crown lengthening (CL), carried out to maintain aesthetics and treat gingival margin discrepancies, can expose tooth structure. CL is usually extended to the adjacent teeth and not limited solely to the targeted tooth for harmonious osseous and gingival contours. However, it may lead to loss of bone support in the adjacent teeth resulting in esthetic concerns such as long clinical crowns, flattened papillae, and black triangles [5,6,7].

Few studies have investigated the placement of a crown on endodontically treated teeth (ETT) [5,6]. There is a lack of evidence supporting its placement over a direct restoration on severely broken down ETT [7]. The final position of the gingival margin post-recovery is affected by factors such as the immediate post-suturing position of flap margin [8], amount of osseous resection [9], the experience of clinicians [10], gingival biotype [8], inter-individual variations of biologic width [11], and post-surgical bone remodeling [8]. Healing time for maturation and stability of periodontal tissue must also be considered before placement of a permanent restoration in the aesthetic areas.

The material and fabrication technique of the indirect restoration plays an important role in its success and longevity [12]. DME facilitates the placement of a large direct composite restoration and is an alternative to surgical CL. Treatment selection may be affected by root concavity, furcation, medical history, and the presence of implants [12,13]. CL technique poses risks of esthetic complications, infections, implant thread exposure, and destabilization of an implant. Indirect impression adhesive restorations can be complicated, as isolation and delivery can be affected by localized subgingival margins that hinder its durability and adhesion with the periodontal tissues.

Debate continues as to whether a non-invasive elevated margin technique or surgical CL is the better strategy facilitating the placement of large direct composite resin restorations. Though a conservative approach is often advocated, it fails in situations that demand change in the shape of tissues around the tooth for restoration [14]. This systematic review examined the survival rate of badly decayed teeth when managed with crown lengthening and compared it to the deep margin elevation technique.

## 2. Materials and Methods

### 2.1. Study Protocol

This systematic review was conducted following the Preferred Reporting Items for Systematic Reviews and Meta-Analyses (PRISMA) guidelines (17–19). The following focused question was developed in accordance with the PICO format:

‘Does the crown lengthening technique (I) provide a better survival rate (O) than deep margin elevation technique (C) following the restoration of badly decayed teeth (P)?’

### 2.2. Eligibility Criteria

#### 2.2.1. Inclusion Criteria


Clinical trial studies with a follow-up period of more than six months.Studies reporting the baseline and post-treatment measurements.Clinical measurements were obtained through the reference method.Studies published in the English language.


#### 2.2.2. Exclusion Criteria


Case reports, editorials, and clinical letters.Studies without subsequent restorative needs were also excluded on account of the emblematic differences in typical surgical sites and etiology.


### 2.3. Search Strategy and Study Selection

The search was conducted during July 2020 and then updated at the end of July 2021, and no restriction concerning publication status and time was applied during the search. Cochrane Database, EBSCO, Scopus, and Medline databases were searched electronically for relevant literature. Google Scholar was used as a secondary source. Three authors (S.B., Z.K., and S.J.) independently performed the searches using the following keywords and Boolean operators: ‘surgical crown lengthening’, ‘crown lengthening’, ‘deep margin elevation’, ‘decayed tooth’, ‘prognosis’, ‘indirect adhesive restorations’, ‘composite resin base’, ‘anatomical complications’, ‘supragingival elevation’, ‘rubber dam’, ‘gingival disease’, ‘periodontal attachment loss’, and ‘short clinical crowns’. Duplicates were removed manually, following which the titles and abstracts of the articles were screened independently by three reviewers based on predefined inclusion and exclusion criteria. The full text was reviewed if additional information was required. Any discrepancy was resolved through discussion and consensus with a fourth author (A.T.R.). Later, A.T.R. and S.J. reviewed the full text of all the remaining articles for eligible studies. Later, a manual search of the references in the selected articles was conducted to identify any additional articles.

### 2.4. Quality Assessment

The methodological quality of each study was assessed using relevant guidelines from the Cochrane Handbook for Systematic Reviews [15]. Six domains were assessed based on the reporting of randomization, allocation concealment, blinding, incomplete outcome data, selective reporting, and other sources of bias. Two independent reviewers (M.E.S. and B.M.N.) assessed the quality of the studies, and any discrepancies in ratings were resolved through discussion with a third author (M.H.M.) until a consensus was reached.

### 2.5. Data Extraction and Synthesis

Relevant data were extracted from the selected articles individually by two reviewers (S.J. and Z.K.) using separate customized charts. Relevant information such as the name of the author, year of publication, sample size, mean age, observation time, and all details related to CL and DME were entered into the charts.

## 3. Results

### 3.1. Study Selection 

A total of 5269 articles were found from the electronic database search. Overall, 1042 duplicates were removed and 4201 articles were excluded on the basis of the title and abstract. The full text of 26 articles was examined for eligibility. Based on the inclusion and exclusion criteria, 6 articles were selected for this review. Figure 1 depicts the flow chart of the study selection.

### 3.2. Quality Assessment

Of the six studies selected in this review, five showed a high risk of bias. The five articles showed a lack of randomization and allocation concealment. There were serious concerns regarding blinding and incomplete data. Thus, the overall risk of bias was high. Figure 2 depicts the summary of the quality assessment.

### 3.3. Characteristics of the Studies

A total of six studies published between 1992 and 2020 were selected for this review. The characteristics of the selected studies are summarized in Table 1, including the type of the study, study year, age of the participants, time of follow-up, and CL from baseline to follow-up, and free gingival margin position from baseline to follow-up period.

None of the selected studies reported on the pre-surgery measurement of CL for the treated teeth or control. Mean CL change was determined from the recorded free gingival marginal (FGM) position change. The average value at 6 months varied from 1.4 mm [16], 3.33 mm [17], 1.72 mm [8], and 1.57 mm [18]. In one study, CL remained stable postoperatively [16], while in two investigations [8,18,19,20,21] the CL decreased during follow-up (*p* < 0.05). One study, however, did not report immediate postoperative measurements [22]. Most treated teeth were reestablished by the study’s end [16,17,18]. None of the studies examined whether the accomplished CL was satisfactory for restorative treatment without operational correction.

All studies utilized an apically positioned flap (APF) and measured osseous/bony resection. Flap margins were positioned at the crest level in one study [17]. One study [17] utilized an aggressive bony resection, where 90% of the treated sites went through osteotomy ≥3 mm. Other studies followed a moderate approach: 94% of treatment sites had osseous resection ranging from 1–3 mm [8], 93.3% had osseous resection ≤2 mm [4], and 96% had osseous resection ≤2 mm [16] of the treated sites.

Three studies reported the immediate clinical outcomes of CL surgery at the end of suturing [8,16,18]. FGM levels were calculated to evaluate the CL increment immediately after surgery. A significant mean apical recession was noted. It varied from 1.32 mm (*p* ≤ 0.05) [16] to 2.27 mm (*p* < 0.005) [18] and 2.50 mm [8] in the studies.

Long-term periodontal outcomes were assessed at six or after. The inflammation/plaque parameters (BOP, GI, PL, and PI) were significantly different from baseline assessments [8,18]. PD showed a significant reduction from baseline to six months post-surgery (*p* < 0.05). Three studies found CAL to be significantly different in relation to baseline (*p* ≤ 0.05 [21], *p* < 0.001 [17]).

In the long-term outcome, three studies [16,17,18] found that the FGM position was significantly displaced apically in comparison with the baseline at the sites of treated (TT), not adjacent (NA), and adjacent (AA). These findings were unchanged when compared with the control group [16]. FGM position change at six months and baseline was more prominent at TT compared to AT and NA sites [8,17,18]. Similar to the CL outcomes, the average FGM on TT was 0.78 mm compared to 0.70 mm [18] coronal to the immediate post-surgery position. The summary of the studies for CL surgery are presented in Table 1.

**Table 1 materials-14-06733-t001:** Summary table of studies for CL surgery.

Study Year	Brägger et al. [21] 1992		Lanning et al. [22] 2003		Arora et al. [11] 2013		Deas et al. [23] 2004	
Study type	Clinical analysis		Clinical Evaluation		Clinical study		Clinical study	
Sample size (Gender)	25 (NAD) 43 test and 42 control teeth		23 18 (NAD)		64 patients (M = 38 F = 26)		25 patients (M = 18 F = 7)	
Mean Age (range)	20–81 years (61)		28–72 years (39)		18–63 (34.5)		(NAD)	
Crown length after surgery (mm)	TT Control	1.3 -	TT AA NA	- - -	TT AA NA	2.5 + 0.72 1.89 + 0.83 1.18 + 0.73	TT AA NA	2.27 + 1.10 2.18 + 0.98 1.06 + 1.07
Change in crown length after 6 months follow up (mm)	TT Control	1.4 0.2	TT AA NA	3.33 + 0.15 2.82 + 0.24 2.78 + 0.28	TT AA NA	1.72 + 0.80 1.41 + 0.88 0.72 + 0.58	TT AA NA	1.69 + 1.01 1.43 + 0.96 0.84 + 0.85
*p*-value			< 0.05 (for TT)		< 0.05		< 0.005	
Baseline Free gingival margin position (mm)	TT Control	6.02 + 2.39 5.83 + 2.15	TT AA NA	4.57 + 0.29 4.65 + 0.24 4.51 + 0.26	TT AA NA	-	TT AA NA	-
After surgery gingival margin position (mm)	TT Control	7.34 + 2.30 -	TT AA NA	- -	TT AA NA		TT AA NA	2.27 + 1.10 2.18 + 0.98 1.06 + 1.07
*p* value (baseline and after surgery)	*p* ≤ 0.05							
After 6 months follow up gingival margin position (mm)	TT Control	7.39 + 2.00 6.01 + 1.93	TT AA NA	3.33 + 0.15 2.82 + 0.24 2.78 + 0.28	TT AA NA	1.72 + 0.11 1.41 + 0.12 0.72 + 0.08	TT AA NA	1.57 + 1.01 1.30 + 0.96 0.76 + 0.85
*p*-value (baseline and after 6 months)	TT	*p* ≤ 0.05	TT AA NA	*p <* 0.0001 *p <* 0.0001 *p <* 0.0001			TT AA NA	*p <* 0.005 *p <* 0.005 *p <* 0.005
BW and SGT at baseline (mm)	TT Control	-	TT AA NA	2.26 + 0.13 2.23 ± 0.11 2.36 ± 0.08	TT AA NA	3.50 ± 0.83 3.73 ± 0.82 3.62 ± 0.64	TT AA NA	-
BW and SGT at 6 months (mm)	TT Control	-	TT AA NA	−0.07 ± 0.09 −0.15 ± 0.07 −0.23 ± 0.06	TT AA NA	−0.44 ± 0.12 −0.30 ± 0.12 −0.48 ± 0.08	TT AA NA	-
*p*-value (baseline and after 6 months)			TT (not different from baseline) AA (*p <* 0.05) Na (*p <* 0.05)		TT (*p =* 0.001) AA (*p =* 0.049) NA (*p =* 0.000)			
Bone level changes at 6 months (mm)	TT Control	-	TT AA NA	3.50 ± 0.59 2.54 ± 0.76 2.47 ± 0.88	TT AA NA	1.27 ± 0.10 1.11 ± 0.08 0.23 ± 0.06	TT AA NA	-
*p*-value (from baseline and 6 months)			TT (*p <* 0.0001) AA (*p <* 0.0001) NA (*p <* 0.0001)		TT AA NA		(ND)	

Non-adjacent cite (NA), Adjacent cite (AA), Treated site (TT), Biologic width (BW), Supracrestal gingival tissue (SGT), Statistical significance not documented (ND), Not available data = NAD.

The connection between tissue rebound and flap position post-suturing was measured. An inverse connection was seen where the flap edges sutured nearer to the osseous crest showed prominence (r = 0.601, *p* = 0.000) (v); r = 0.422, *p* < 0.01 [18]. When the immediate post-surgical distance between the osseous crest and flap position at TT was greater than 2 mm, long-term tissue rebound was average (≤ 0.50 mm) [3] and ≤ 0.47 mm [18]. When this distance was less than 1 mm, a greater tissue rebound ranging from 1.33 mm to 1.42 mm was observed [8,18]. This connection between flap position post-suturing and tissue rebound was visible at TT, AA, and NA sites [8,18] and interproximal and lingual sites [18].

Moreover, a statistically significant correlation (*p* < 0.001) was found between periodontal biotype and tissue rebound [8]. A higher rebound was due to a thicker biotype (r = 0.325). In the long-term, thin scalloped biotype and flat thick leveled biotype had an average of 0.37 and 0.70 mm of tissue rebound. Tissue rebound was found to be the same regardless of the type of tooth treated, the anatomical anteroposterior position, or whether they were treated separately or in a group [8,18]. The differences in tissue rebound between interproximal and lingual sites were insignificant [18].

Long-term BW [17] and SGT [8] measurements at TT were smaller when compared with baseline. SGT changes were measurably significant (*p* = 0.001). Bone changes at baseline and six months were measured in two studies [8,17] where the apical displacement was average 3.50 mm at TT (*p* < 0.0001) [17] and 1.27 mm [8]. Compared to NA and AA, the TT site had more significant apical displacement (*p* < 0.05) [17]. None of the studies reported any post-surgical complications in patient outcomes (patient-reported findings) [8,16,17,18].

Two studies [19,20] examined outcomes of DME and their results were heterogeneous. The details of the studies are summarized in Table 2. In the first study [19] eight out of 189 restorations failed within 3.8 to 4.7 years, with a single case each of fracture, severe periodontal breakdown, pulpal necrosis, and five cases of secondary caries. The results revealed an overall 95.9% success in survival over the period of ten or more years. Older restorations had higher fracture rate of tooth itself (*p* = 0.000) and indirect restorations (*p* = 0.000), had higher margin discoloration (*p* = 0.01) and more caries (*p* = 0.000). Fractures of teeth (*p* = 0.000) and restorations (*p* = 0.000) were present in cases that had a history of endodontic treatment. All other predefined variables did not have a statistically significant influence on survival or quality of treatment (*p* > 0.05). The summary of the studies for DME are presented in Table 1.

In the second study [20], all samples survived the aging test. Fracture strength was not significantly influenced by the DME (*p* = 0.15). Onlays with DME had greater fracture strength than inlays with DME (*p* = 0.001). However, inlays fractures were more repairable than onlays. Fractures were classified into five types with their reparability (Table 2). The reparability (*p* < 0.001) and the type of fracture (*p* < 0.001) were significantly affected by the preparation design and not by DME (*p* > 0.05).

## 4. Discussion

This systematic review focused on the prognosis of a severely decayed tooth treated with crown lengthening surgery (CL) versus deep margin elevation (DME). A total of six studies that had a minimum follow-up time of 6 months were selected. The clinical indices in all studies confirmed that all patients were free from comorbidity and acquiescent in maintaining oral hygiene. The information regarding BW or osseous crest position was given only in two trials.

For CL, various clinical outcomes were assessed in both interventions, and a baseline value was utilized for clinical estimations. Post-surgery there was an increase in CL. However, that increased length was found to be significantly reduced at the follow-up [8,18]. The free gingival margin (FGM) displayed significant displacement while healing, but showed overall stability at 6 months [16]. Most of this displacement occurs within the first six months of post-surgery. These results are consistent with data obtained by Lanning et al. who did not observe major changes in the gingival margin position over 6 months [17]. However, Pontoriero et al. found significant changes in the gingival margin over a longer follow-up (12 months) [21]. A possible explanation of this could be a different healing response among different biotypes and sites (interproximal/buccal/lingual) [22].

From the limited data available, it appears that that prognosis after CL surgery is dependent on anatomical and technical factors. The post-surgical maturation and healing in CL of the periodontal tissue involve bone remodeling in density with changes in the soft tissue (such as regrowth, recession, or stability) and corresponding crest height resorption [23]. Changes to the gingival margin position ensue glacially over the healing period. This would extend the time taken for the final restoration to be delivered. The final restoration post-CL could be carried out after 6–12 weeks on posterior areas and 3–6 months on the anterior area of teeth [24]. Even though surgical and biological factors are essential for the healing process, esthetic concerns are the main clinical parameter that influences this decision. Based on the available evidence, the cost of maintenance, and the long-term survival of retained tooth favors tooth over implants CL surgery appears successful for the long-term retention of teeth [25,26].

DME, in conjunction with indirect restorations, has a better survival ratio compared to CL. Similarly, restorations on non-vital teeth and composite resin indirect restorations also show survivability with DME. However, the initial survival rate can degrade over time. In one study, DME did not influence the fracture strength of ceramic restorations [20]. However, data suggests that the fracture strength is similar when measured with or without DME. These results coincide with a previous study that examined fracture strength when teeth are restored with DME [27]. Oblique forces, rather than bite force, are causative in the onlays or inlays fracture, with or without DME. Therefore, both onlays and inlays are clinically fracture resistant, regardless of DME.

A higher failure rate was observed in indirect restoration made from composite compared to ceramic [24]. More degradation was observed in older restorations (greater than 3 years). Failure includes discolored margins, fractured teeth, and restoration, and an increase in caries rate. These failures could be attributed to the presence of pre-existing fractures and fissures in the remaining cusp, and the presence of pre-existing amalgam restoration [24,25]. No decline in periodontal health was reported when DME was performed [24].

A recent case report, evaluated CL vs. DME to help in clinical decision making and suggested DME as a better alternative to CL for deep cavities [14]. However, this conclusion was based on the outcome of biological width and not on the survival ratio or successful retention. Nevertheless, based on the limited evidence available, their findings are in line with the conclusion of this review. There is an absence of evidence to support the effectiveness of CL over DME for the restoration of severely decayed teeth. DME is clinically and histologically well tolerated by the surrounding periodontium. Further research with long-term follow-up is required to confirm and validate these findings.

### 4.1. Completeness and Applicability of Evidence

Although six studies were included in this review, only a small number of studies were eligible for each comparison. Studies did not report on specific patient-centered outcomes or responses from the restorative dentist regarding the clinical outcomes post-crown lengthening. It remains unclear whether the post-surgical CL results were adequate for the planned restorative needs without a revision procedure. None of the studies examined the mobility of the teeth post-surgery or the crown-to-root ratio outcomes which may have a bearing on the success of the restoration [28]. Few studies have reported long-term follow-up post-CL surgery longer than six months. Further investigation is needed on the outcomes of CL surgery in terms of the entire tooth versus the involved tooth part with surface-specific data. Meta-analysis was not conducted due to clinical heterogeneity in terms of the patient population, study design, and outcome measures among the studies. The findings of this review must be interpreted with caution, as a majority of the studies were found to have a high risk of bias.

### 4.2. Quality of Evidence

Only one of the studies included in this review was randomized and was assessed to have a low risk of bias. The remaining five studies showed a high risk of bias due to insufficiencies in the study design due to lack of randomization, blinding of investigators, and outcome assessors. Additionally, missing data due to losses during follow-up were indicators of a high risk of bias. Overall, this resulted in a lower certainty for the evidence. The paucity of high-quality trials evaluating CL proved disappointing as this leads to statistical limitations when meta-analysis is attempted. The most common problem is the lack of randomization and study design. With regard to blinding, operators and patients would be aware of the intervention and procedures. It is conceivable that blinding may be challenging considering the nature of the intervention. The generalization of the results should be considered with caution.

### 4.3. Limitations

Every effort was made to limit the potential for any biases in the review by conducting a comprehensive search and gathering multiple authors’ independent assessments in deciding the eligibility of studies for inclusion. This minimized the selection bias. Comprehensive search protocol and constant quality checks were the strengths of this systematic review. Multiple authors independently conducted the data extraction. One limitation is that only studies in the English language were considered.

## 5. Conclusions

Within the limited available evidence, crown lengthening (CL) surgery is successful for long-term retention of teeth. However, deep margin elevation (DME) has a better survival ratio. This review revealed that there is a dearth of high-quality trials examining prognosis following the restoration of badly decayed teeth with deep margin elevation versus crown lengthening. Further studies are needed to take follow-up over a longer time to assess the rebound effect of CL surgery over time and degradation in DME. Future investigations should also consider patient-reported outcomes such as pain, discomfort, cost, and satisfaction. Well-designed trials using defined objective measures and detailed high-quality descriptions of methodology should be undertaken to determine the best clinical approach and create guidelines for the treatment of severely decayed teeth. An evidence-based approach is vital for providing patients with the most effective, acceptable, and cost-effective restorations

## Figures and Tables

**Figure 1 materials-14-06733-f001:**
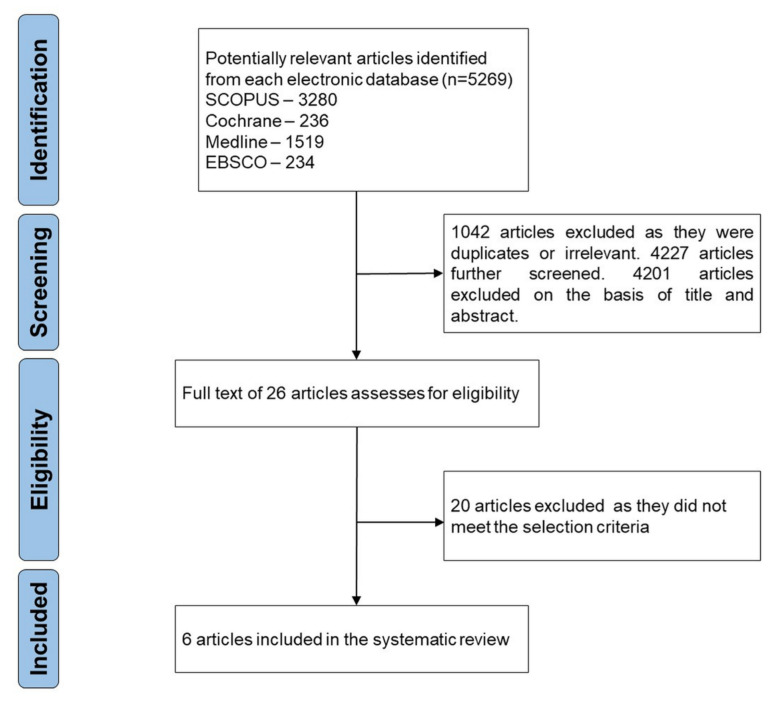
PRISM flow chart of the study selection.

**Figure 2 materials-14-06733-f002:**
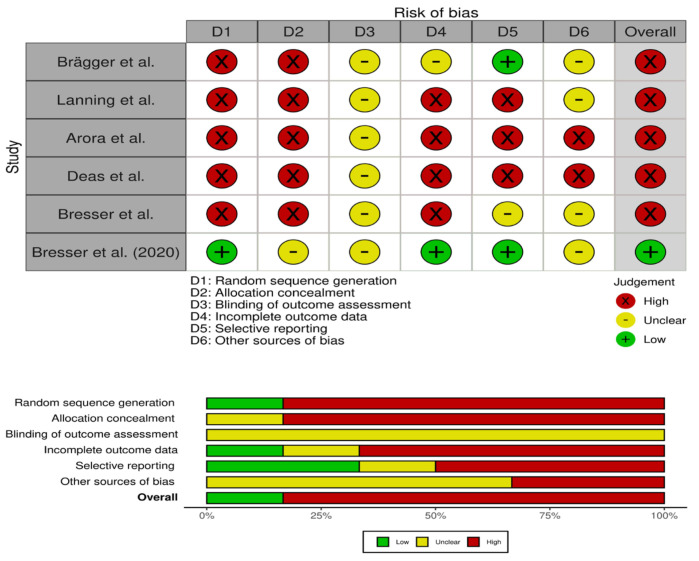
Summary of risk of bias.

**Table 2 materials-14-06733-t002:** Summary table of studies for DME.

Study Year	Bresser et al. [24] 2019		Bresser et al. [25] 2020					
Study type	Clinical study		Clinical Evaluation					
Sample size (Gender)	197 indirect restorations in 120 patients (M = 78, F = 42)		InoD = 15 InWD = 15 OnoD = 15 OnWD = 15 (NAD)					
Age mean (range)	76 years (30–106 years)		-					
Observation time (range)	57.7 months (4–144 months)		-					
Fracture Type N Inlay n (%) Outlay n (%)	-		I	II	III		IV	V
			2	16	6		0	36
			2 (100)	14 (87.5)	5 (83)		0	9 (25)
			0	2 (12.5)	1 (17)		0	27 (75) *p <* 0.0125
Repairability N Inlay n (%) Outlay n (%)	-		Repairable			Irreparable		
			24			36		
			21 (88) (*p <* 0.00625)			9 (25)		
			3 (12)			27 (75) (*p <* 0.00625)		
Failure of treatment	secondary caries (n = 5) pulpal necrosis (n = 1) periodontal breakdown (n = 1) fracture (n = 1),							
Survival rate	95.9% after > 10 years							
USPHS (>3 years versus <3 years old)	more fracture of indirect restoration	χ2(2) = 42.03, *p =* 0.000						
	fracture of the tooth	χ2(2) = 23.18, *p =* 0.000						
	Carries	χ2(2) = 9.02, *p =* 0.000						
	margin discoloration	χ2(2) = 9.02, *p =* 0.01						

Type I = ¼ enamel fracture, type II = ¼ ceramic fracture, type III = ¼ ceramic-enamel fracture, type IV = ¼ ceramic-dentin fracture, type V = ¼ crown-root fracture, inlay without DME = InoD, inlay with DME = InWD, onlay without DME = OnoD, onlay with DME = OnWD, and Not available data = NAD.

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
