# Peer review of "Treatment Prognosis of Restored Teeth with Crown Lengthening vs. Deep Margin Elevation: A Systematic Review"

_materials, 2021, doi:10.3390/ma14216733_

Round 1

Reviewer 1 Report

This is an interesting topic and highly discussed in current Dental Meetings.

I understand the lack of consistent clinical data about DME. Maybe it is too son for a systematic review on which just ONE reference really could be used (ref. 25).

I would recommend you to:

  • focus on this lack in the Discussion.
  • Go deeply in reference 25, discussing the failures observed in the follow-up of the author’s cases.
  • Include more Biological response discussion on DME and how the only study that evaluated clinical results observed the periodontal response (measurements, etc.).

Author Response

Reviewer 1:

Comments

This is an interesting topic and highly discussed in current Dental Meetings.

Thank you for kind words. We are grateful for the encouragement.

I understand the lack of consistent clinical data about DME. Maybe it is too son for a systematic review on which just ONE reference really could be used (ref. 25). focus on this lack in the Discussion.

New text added

Lines 290-296

A higher failure rate was observed in indirect restoration made from composite as compared to ceramic [24]. More degradation was observed in older restorations (greater than 3 years). Failure includes discolored margins, fractured teeth, and restoration, and an increase in caries rate. These failures could be attributed to the presence of pre-existing fractures and fissures in the remaining cusp, and the presence of pre-existing amalgam restoration [24,25]. No decline in periodontal health was reported when DME was performed [24].

Go deeply in reference 25, discussing the failures observed in the follow-up of the author’s cases

Include more Biological response discussion on DME and how the only study that evaluated clinical results observed the periodontal response (measurements, etc.).

Reviewer 2:

ABSTRACTS

The abstract has an agonizing low quality. The authors are not clearly prepared to prepare a manuscript like this, or if are were completely sloppiness when preparing the Abstract.  The 4 sections that are present in the Abstract (Purpose, Methods, Results and Conclusions) must be re-written, I advise reading the PRISMA guidelines in order to do so. The way it is right now is unpublishable.

Abstract has been rewritten:

Crown lengthening surgery and deep margin elevation are two distinct approaches used to manage decayed teeth. This systematic review examined the survival rate of badly decayed teeth when restored using the crown lengthening technique and compared it to the deep margin elevation technique.

An electronic search of Scopus, Medline, EBSCO, Cochrane library, and Google scholar databases were conducted to identify relevant articles published up to July 2020. Predefined inclusion and exclusion criteria were used to select the relevant articles. PRISMA guidelines were followed. The focused PICO question was: ‘Does the crown lengthening technique (I) provide better survival rate (O) than deep margin elevation technique (C) following the restoration of badly decayed teeth (P).’ A total of six articles were included after performing screening based on the eligibility criteria. Four studies focused on crown lengthening while two focused on deep margin elevation technique. A majority of the studies showed a high risk of bias owing to methodological insufficiencies. CL treated cases showed a change in the free gingival margin at six months’ post-surgery. Tissue rebound was seen that was correlated to the periodontal biotype. Teeth treated with DME technique showed high survivability. There is a lack of high-quality trials examining surgical comparisons between CL and DME with long-term follow-up. Patient and dentist reported outcomes have not given adequate consideration in literature. Based on the limited evidence, it can be concluded that for restorative purposes, crown lengthening surgery is successful for long-term retention of teeth. However, deep margin elevation technique has better survival ratio. Future well designed and executed research will have an effect on the evidence and level of certainty for the best approach to treating severely decayed teeth.

KEYWORDS

Add “Systematic Review” to the keywords

Added

Line 51

INTRODUCTION

 The introduction has 2 pages, should be somehow reduced since it is too long.

The introduction has been streamlined

Can you make the rationale of the review more clear?

Aim statement rewritten to make aim and rationale more clear

New (Lines: 96-98)

This systematic review examined the survival rate of badly decayed teeth when managed with crown lengthening and compared it to the deep margin elevation technique.

The aim sentence has to be completely reformulated. The authors state the objective is to “compare the treatment prognosis”, but in which aspect? Which outcome was chosen to classify the prognosis? Can the authors be more clear?

METHODS

In the “Study Protocol”, the sentence “systematic review was conducted considering the basics of Cochrane Handbook for Systematic Reviews of Interventions [17] as stated by the Preferred Reporting Items for Systematic reviews and Meta-Analysis (PRISMA) statement [18,19]. The study was designed, conducted, and reported following quality standards and guidelines for reporting systematic reviews and meta-analysis.” is repeating it self. May the authors state it in a single sentence?

Rewritten:

(Lines: 102-104)

This systematic review was conducted following the Preferred Reporting Items for Systematic Reviews and Meta-Analyses (PRISMA) guidelines.

Please correct the verb to the past in the following sentence: “The review is submitted for registration in 136 PROSPERO”

Statement has been removed.

When was the search conducted? (date: start and finish)

Line 121

Search was conducted in July 2020 and no restriction concerning publication status and time was applied during the search. All published studies until the date were subjected to the Selection criteria.

Did the authors conducted a manual search? This is advisable in all reviews, if they did not I advise to do it now and re-submit the manuscript in another time. Manual search is a very relevant source of information.

Yes. Manual search was conducted as it important to make sure that no articles were missed. Indicated in text:

Line: 133

Later a manual search of the references in the selected articles was conducted to identify any additional articles

Did the authors contact previous authors asking for more studies if available?

No such contact was made

“Quality assessment”, I am not understanding how this step was conducted. Who made this step? 1 or 2 observers? What was the overall agreement between them? The final results were consensual? A lot of information is missing here.

The section has been rewritten

New: (Lines 137-142)

The methodological quality of each study was assessed using relevant guidelines from the Cochrane Handbook for Systematic Reviews [15]. Six domains were assessed based on the reporting of randomization, allocation concealment, blinding, incomplete outcome data, selective reporting, and other sources of bias. Two independent reviewers (M.E.S., B.M.N.) assessed the quality of the studies, and any discrepancies in ratings were resolved through discussion with a third author (M.H.M.) till a consensus was reached.

“Data extraction and synthesis”. Which data was extracted? If you registered in PROSPERO you had this previously defined. What was the outcomes of measure that you were aiming for?

Review was not registered in PROSPERO.

145-148

Relevant data were extracted from the selected articles by two reviewers (S.J. and Z.K.) using a customized chart. Relevant information such as the name of the author, year of publication, sample size, mean age, observation time, and all details related to CL and DME were entered into the charts (Table 1 & 2).

RESULTS

Regarding the flow chart, can you present the “n” according to each source? Specify the source and the n to each one of them.

Done.

Still on the flow chart, you talk about “additional records identified tru other sources”. Which sources were this? May you specify? Were they stated in the methods?

This statement is removed

Regarding the n=16 and n=9 that were excluded (Flow chart), may the authors identify the reasons for exclusion?

Old figure is replaced with new one

Articles were rejected as they did not meet the inclusion criteria.

Similarly to the Introduction, the Results section is too long. There are informations in Tables and the authors should be able to summarize a little better. My recommendation is to reduce, as possible, the length of the results.

The Results section has been streamlined.

DISCUSSION

The debate of the Discussion seems acceptable, but all the technical aspects of the Systematic review are just absence. Once again, just as in the Abstract, the authors are ignoring the guidelines on how to report these type of studies.

Discussion has been rewritten based on reviewer recommendation

Lines 250-338

Need to write overall completeness and applicability of evidence

Section has been added

Lines 305-317

Completeness and applicability of evidence

Although six studies were included in this review, only a small number of studies were eligible for each comparison. Studies did not report on specific-patient centered outcomes or responses from the restorative dentist regarding the clinical outcomes post-crown lengthening. It remains unclear whether the post-surgical CL results were adequate for the planned restorative needs without a revision procedure. None of the studies examined the mobility of the teeth post-surgery or the crown-to-root ratio outcomes which may have a bearing on the success of the restoration [28]. Few studies have reported long-term follow-up post-CL surgery longer than six months. Further investigation is needed on the outcomes of CL surgery in terms of the entire tooth versus the involved tooth part with surface specific data. Meta-analysis was not conducted due to clinical heterogeneity in terms of patient population, study design, outcome measures among the studies. The findings of this review must be interpreted with caution as a majority of the studies were found to have a high risk of bias.

Need to discuss about the quality of included studies and debate Risk of Bias

Section has been added

Lines 320-330

Quality of evidence

Only one of the studies included in this review was randomized and was assessed to have a low risk of bias. The remaining five studies showed a high risk of bias due to insufficiencies in the study design due to lack of randomization, blinding of investigators, and outcome assessors. Additionally, missing data due to losses during follow-up were indicators of a high risk of bias. Overall, this resulted in a lower certainty for the evidence. The paucity of high-quality trials evaluating CL proved disappointing as this leads to statistical limitations when meta-analysis is attempted. The most common problem being lack of randomisation and study design. In regards to blinding, operators and patients would be aware of the intervention and procedures. It is conceivable that blinding may be challenging considering the nature of the intervention. The generalisation of the results should be considered with caution.

Need to add limitations at study and outcome level and at review level

Section has been added

Lines 333-338

Every effort was made limit the potential for any biases in the review by conducting a comprehensive search and gathering multiple authors’ independent assessment in deciding the eligibility of studies for inclusion. This minimized the selection bias. Comprehensive search protocol and constant quality check were the strengths of this systematic review. Multiple authors independently conducted the data extraction. One limitation is that only studies in the English language were considered.

Please debate Review level of evidence

Done

Need to debate generalization of the findings

Done

Please debate review strength

Done

New lines: 335-337

Comprehensive search protocol and constant quality check were the strengths of this systematic review. Multiple authors independently conducted the data extraction.

REFERENCES

Some references have the papers titles in capital letters, others do not. Please review all references in order to make them standard.

Done

Reviewer 3

Provide the definitive PROSPERO registration number as soon as available (after 30 days it should be available)

Study was not registered in PROSPERO

Formulate the Pico Question

Done

New Lines 105-106

 ‘Does the crown lengthening technique (I) provide better survival rate (O) than deep margin elevation technique (C) following the restoration of badly decayed teeth (P)?’

Indicate the Records identified for each database for each keyword or combination of words and provide them as supplementary materials

Done

Indicate the start and end date of the study search (must coincide within the limits with the one declared in Prospero)

Search was conducted in July 2020 and no restriction concerning publication status and time was applied during the search. All published studies until the date were subjected to the Selection criteria.

Specify the number of authors who searched for articles.

New: (Lines 123-134)

Three authors (S.B., Z.K. & S.J.) independently performed the searches using the following keywords and Boolean operators:‘surgical crown lengthening’, ‘crown lengthening’, ‘deep margin elevation, ‘decayed tooth’, ‘prognosis’ ‘indirect adhesive restorations’, ‘composite resin base’, ‘anatomical complications’, supragingival elevation’, ‘rubber dam’, ‘gingival disease’, ‘periodontal attachment loss, and ‘short clinical crowns.’ Duplicates were removed following which the titles and abstracts of the articles were screened independently by three reviewers based on predefined inclusion and exclusion criteria. Full-text was reviewed if additional information was required. Any discrepancy was resolved through discussion and consensus with a fourth author (A.T.R.). Later A.T.R. and S.J. reviewed the full text of all the remaining articles for eligible studies. Later a manual search of the references in the selected articles was conducted to identify any additional articles. 

Indicate the method for removing overlaps (manual or \ and by software)

Manual method was used to remove duplicates

Line 128

Provide more information to the tables and figures especially the symbols used and Acronyms.

-Done

Acronyms explained

According to the Prisma, to reduce the risk of error in data extraction, this operation should be performed by 2 operators independently and in different tables. Indicate at least one operator if there has been one who has had the task of verifying the data

New: (Lines 176-192)

Relevant data were extracted from the selected articles individually by two reviewers (S.J. and Z.K.) using separate customized charts.

Improve the graphic aspect of tables 1 and 2

Done

Indicate clearly the limits of the study

Done

Section has been added

Lines 333-338

Every effort was made limit the potential for any biases in the review by conducting a comprehensive search and gathering multiple authors’ independent assessment in deciding the eligibility of studies for inclusion. This minimized the selection bias. Comprehensive search protocol and constant quality check were the strengths of this systematic review. Multiple authors independently conducted the data extraction. One limitation is that only studies in the English language were considered.

Reviewer 2 Report

The authors are to be congratulated for all the efforts employed when conducting the present research.

Here goes a few concerns and comments:

ABSTRACTS

The abstract has an agonizing low quality. The authors are not clearly prepared to prepare a manuscript like this, or if are were completely sloppiness when preparing the Abstract.  The 4 sections that are present in the Abstract (Purpose, Methods, Results and Conclusions) must be re-written, I advise reading the PRISMA guidelines in order to do so. The way it is right now is unpublishable.

KEYWORDS

Add “Systematic Review” to the keywords

INTRODUCTION

The introduction has 2 pages, should be somehow reduced since it is too long.

Can you make the rationale of the review more clear?

The aim sentence has to be completely reformulated. The authors state the objective is to “compare the treatment prognosis”, but in which aspect? Which outcome was chosen to classify the prognosis? Can the authors be more clear?

METHODS

In the “Study Protocol”, the sentence “systematic review was conducted considering the basics of Cochrane Handbook for Systematic Reviews of Interventions [17] as stated by the Preferred Reporting Items for Systematic reviews and Meta-Analysis (PRISMA) statement [18,19]. The study was designed, conducted, and reported following quality standards and guidelines for reporting systematic reviews and meta-analysis.” is repeating it self. May the authors state it in a single sentence?

Please correct the verb to the past in the following sentence: “The review is submitted for registration in 136 PROSPERO”

When was the search conducted? (date: start and finish)

Did the authors conducted a manual search? This is advisable in all reviews, if they did not I advise to do it now and re-submit the manuscript in another time. Manual search is a very relevant source of information.

Did the authors contact previous authors asking for more studies if available?

“Quality assessment”, I am not understanding how this step was conducted. Who made this step? 1 or 2 observers? What was the overall agreement between them? The final results were consensual? A lot of information is missing here.

“Data extraction and synthesis”. Which data was extracted? If you registered in PROSPERO you had this previously defined. What was the outcomes of measure that you were aiming for?

RESULTS

Regarding the flow chart, can you present the “n” according to each source? Specify the source and the n to each one of them.

Still on the flow chart, you talk about “additional records identified tru other sources”. Which sources were this? May you specify? Were they stated in the methods?

Regarding the n=16 and n=9 that were excluded (Flow chart), may the authors identify the reasons for exclusion?

Similarly to the Introduction, the Results section is too long. There are informations in Tables and the authors should be able to summarize a little better. My recommendation is to reduce, as possible, the length of the results.

 DISCUSSION

The debate of the Discussion seems acceptable, but all the technical aspects of the Systematic review are just absence. Once again, just as in the Abstract, the authors are ignoring the guidelines on how to report these type of studies.

Need to write overall completeness and applicability of evidence

Need to discuss about the quality of included studies and debate Risk of Bias

Need to add limitations at study and outcome level and at review level

Please debate Review level of evidence

Need to debate generalization of the findings

Please debate review strength

REFERENCES

Some references have the papers titles in capital letters, others do not. Please review all references in order to make them standard.

Author Response

(The authors gave the same response as above.)

Reviewer 3 Report

The topic concerning the elongation of the clinical crown and the repositioning of the margin is interesting. However there are some aspects of the systematic review to be clarified and there was no direct comparison between the 2 techniques.

The review presents the following points that need to be addressed.

  1. Provide the definitive PROSPERO registration number as soon as available (after 30 days it should be available)
  2. Formulate the Pico Question
  3. Indicate the Records identified for each database for each keyword or combination of words and provide them as supplementar materials
  4. Indicate the start and end date of the study search (must coincide within the limits with the one declared in Prospero)
  5. Specify the number of authors who searched for articles.
  6. Indicate the method for removing overlaps (manual or \ and by software)
  7. Provide more information to the tables and figures especially the symbols used and Acronyms.
  8. According to the Prisma, to reduce the risk of error in data extraction, this operation should be performed by 2 operators independently and in different tables. Indicate at least one operator if there has been one who has had the task of verifying the data
  9. Improve the graphic aspect of tables 1 and 2
  10. Indicate clearly the limits of the study

Author Response

(The authors gave the same response as above.)

Round 2

Reviewer 2 Report

Dear authors did well in the Revision. Make sure you explain in the Abstract the meaning of CL and DME. Thank you

Author Response

Reviewer 2:

Dear authors did well in the Revision. Make sure you explain in the Abstract the meaning of CL and DME. Thank you

Reply: As advised, the full forms are added as follows:

Crown lengthening (CL) treated cases showed a change in the free gingival margin at six months post-surgery. A tissue rebound was seen that was correlated to the periodontal biotype. Teeth treated with the deep margin elevation (DME) technique showed high survivability.
